# In Search of Maximal Citizenship in Educational Policy for Young People: Analysing Citizenship in Finnish Religious Education in View of the "Maximal" Conception

**Gabriel O. Adebayo**

Department of Education, Faculty of Educational Sciences, University of Helsinki, 00014 Helsinki, Finland; gabriel.adebayo@helsinki.fi

**Abstract:** The place of religion and how it should be employed in education for citizenship is currently an issue in Europe. The challenges of increasing diversity are the underlying factors. The conception of maximal citizenship (a critical model of citizenship) gives a significant framework for analysis and scholarly perspectives about several European contexts on this matter. However, there is hardly maximal citizenship in Finnish contexts in scholarship. Hence, this work searches for the elements of maximal citizenship in educational policy for young people by employing the policy relating to citizenship in Finnish religious education (RE). Focusing on grades 7–9 of basic education, its primary data is based on selected national policy documents. The data were analysed using critical discourse analysis. The main findings suggest that citizenship in Finnish RE is only somewhat compatible with the characteristics of maximal citizenship. This reveals some policy shortcomings that could negatively affect the potential of critical-mindedness of young people and equal opportunities in a democracy. Hence, some suggestions that could improve the situation are embedded in the paper. Nevertheless, a linguistic conception of citizenship in Finland vis-à-vis a recent development in national educational policy seems to push the conception of maximal citizenship in a relatively new direction. Furthermore, an explicit use of the "Convention on the Rights of the Child" in Finnish curriculum broadens our conception of maximal citizenship in general. Moreover, while scholars generally agree that maximal citizenship is essentially "critical", this piece suggests that every "critical" approach to citizenship education is not necessarily "maximal".

**Keywords:** citizenship education; curriculum; democracy; educational policy; human rights; maximal citizenship; religious education; young people

---

## 1. Introduction

### 1.1. The Topic: Maximal Citizenship and Religious Education (RE)

Education for citizenship could be used to stimulate and enhance young people's engagement with the political process or to promote social cohesion in a diverse society (Jackson 2007; European Commission (EC)—EC/Eurydice 2017). The notion of citizenship education is hardly separable from religious beliefs. Citizenship and religious education (RE) remain inseparable because religion has impacted, is impacting and will continue to influence the behaviour of private citizens and the policies and laws of many non-Western and Western countries, including Finland (Adebayo and Mansikka 2018; Fouts and Lee 2005). The impact of religion on private citizens and the State could be seen in

their behaviour and attitudes towards non-secularization, secularization and post-secularization in the spheres of private and institutional religion (Poulter 2017).[1]

The need to give relevant citizenship education to young people seems to have prompted the Council of Europe (CoE) and the EC to sponsor research projects on citizenship: Education for Democratic Citizenship (CoE project—EDC) and the Eurydice project on citizenship (EC project) (Jackson 2007). Following Jackson (2007), the findings of these transnational bodies' projects bear out the importance of McLaughlin (1992) theory of maximal citizenship as against its alternative, minimal citizenship. Maximal citizenship could be defined as a model of citizenship that promotes critical understanding and continual questioning about civic matters among citizens. It transcends the uncritical socialization promoted in minimal citizenship (McLaughlin 1992). Accordingly, there is considerable support in Europe for a more maximal interpretation of citizenship in matters of curricula and pedagogy in schools (Jackson 2007, pp. 30–32; cf. EC/Eurydice 2017).[2]

McLaughlin's maximal citizenship approach promotes education about religious beliefs in public or liberal education with a view to promoting justice and producing critical citizens in a democratic and diverse society (McLaughlin 1992). Accordingly, maximal citizenship in RE focuses on the need to promote critical-mindedness for social transformation and justice in religiously and culturally diverse societies. It is currently a growing research interest in some democratic contexts, e.g., English and Dutch (Castelli and Trevathan 2008; Miedema and Avest 2011; Pike 2012; Miedema and Bertram-Troost 2014; Waghid 2014).

Castelli and Trevathan (2008) use McLaughlin (1992) minimalist-maximalist theory to analyse the nature and problem of English citizenship and English Islam and their impact on children's moral, spiritual and educational development. Their primary concern is inadequate engagement between public civic life and the RE of the minorities (notably Islamic RE) regarding citizenship education. According to them, uncritical socialization and unexamined political values are unavoidable consequences of this development. This can "lead to an imposition of unexamined morality that does not take account of the myriad of complexities in any given situation resulting in the imposition of social norms, rather than the development of moral reasoning" (Castelli and Trevathan 2008, p. 88).

Miedema and Avest (2011) and Miedema (n.d.) also lean on the minimalist-maximalist theory of McLaughlin (1992). They suggest that Dutch students' personal religious identity could be developed and strengthened through the combination of citizenship education and RE. They see RE and personal religious identity development as part of citizenship education. For them, this is possible through maximal interpretations of citizenship education in which every student can fashion their identity and worldview through encounter and dialogue in interreligious classes in state schools. This negates the minimal approach that could subtly impose worldviews and identity on students (Miedema n.d., pp. 1–3; Miedema and Avest 2011, pp. 410–15). Miedema (n.d.) suggests that the same principles are also applicable to faith schools provided they (1) interpret RE as part of personal identity development and (2) combine this with a transformative pedagogy that emphasizes the actorship and authorship of the students (Miedema n.d.; Miedema and Avest 2011, p. 414). For Miedema and Avest (2011), citizenship education and RE could and should be fruitfully combined by using a maximal perspective in this secular age.

Pike believes that the transformative aims of citizenship curriculum are in McLaughlin's maximal not minimal interpretation (Pike 2012, p. 110). He regards RE as a necessary and distinctive contributor to citizenship education (Pike 2012; cf. Pike 2008). He argues that citizenship and RE (in England) should facilitate reflection on the beliefs and assumptions underpinning human modes of living. This is to enable learners to explore different interpretations as opposed to "legitimating an uncritical allegiance to secular, liberal and democratic values" (Pike 2012, p. 110). He is critical of any liberal

---

[1]  Meanwhile, the debates relating to the impact of religion on citizenship vis-à-vis non-secularization, secularization and post-secularization are not the main concerns of this paper.

[2]  Note that minimal and maximal citizenship are not necessarily opposite to each other (see more below).

viewpoints promoting exclusion of religious viewpoints in citizenship education. For him, citizenship education founded on liberalism is not neutral (contrary to its often-touted neutrality). He notes that liberalism teaches people to "believe" and "subscribe" to the values of a pre-determined liberal democracy at the least (Pike 2012, pp. 109–11). Hence, RE and citizenship lessons should help young people examine how religious practices are perceived. The lessons should also "encourage them to look at different religious responses to liberal and secular values" (Pike 2012, p. 111, cf. pp. 117–18) in order to realise a balanced perspective.

Miedema and Bertram-Troost (2014, pp. 73–79) argue that the state and the schools have complementary responsibilities to ensure that students are acquainted with and engaged with diverse societal viewpoints. Hence, they advocate a formal education that will reconcile democratic citizenship education, worldview (interreligious inclusive) education and human rights education. Miedema and Bertram-Troost (2014, p. 74) suggested that this "tripartite intertwinement" could be strengthened with McLaughlin (1992) maximal citizenship education. For them, this will meet the communal, liberal and justice demands of citizenship amidst diversity as it emphasizes active learning and inclusion with critical, dialogical and interactive perspectives (Miedema and Bertram-Troost 2014, p. 75).

Waghid (2014) examined three interrelated Islamic educational concepts: *tarbiyya* (nurturing/rearing), *ta'lim* (learning), and *ta'dib* (goodness) in relation to McLaughlin (1992) minimalist-maximalist interpretation. He observed that Islamic-associated concepts do not have a given single meaning but rather meanings that are being shaped by the minimalist and maximalist conditions constituting such concepts. Therefore, he conceived of Islamic education in accord with liberal and democratic citizenship education.

Accordingly, a maximalist interpretation of *tarbiyya* requires students to critically query the basis of Islamic tenets. This goes beyond the minimalist position that would uncritically accept such tenets as unquestionable sacred norms (Waghid 2014, pp. 335–36). A maximalist account of *ta'lim* calls for deliberations on issues that might arise in learning (Waghid 2014, p. 337). This goes beyond the minimalist approach that would simply encourage memorization of Quranic texts, for instance (Waghid 2014, p. 336). In the maximalist view of *ta'dib*, every person deserves respect. This educational view demands justice for all people. It also holds that the ownership of goodness is not the reserved property of a specific group of persons—Muslims or non-Muslims (Waghid 2014, p. 337). On the contrary, a minimalist view of *ta'dib* places the Muslim community above others (Waghid 2014, p. 337).

### 1.2. Citizenship in Finnish RE and Maximal Deficiency

The above suggests that there is a growing research interest in maximal citizenship and RE in Europe, notably in England and the Netherlands. However, an examination of maximal citizenship in Finnish RE (RE is formally known as "Religion" in the Finnish National Core Curriculum for Basic Education—NCCBE) seems to be generally lacking. To begin with, "the civic argument has not been explicitly used to justify the place of RE in state schools and the study of RE in Finland" in the current debate (Poulter 2017, p. 190). Although citizenship education has been treated in many studies in Finland (e.g., Piattoeva 2009, 2010a; Torney-Purta 2002), little concern is being given to the relationship between citizenship and Finnish RE. Besides Poulter (2013, 2017) and Adebayo and Mansikka (2018), one hardly finds any studies that specifically deal with citizenship in Finnish RE, let alone maximal citizenship in Finnish RE; hence, this inquiry.

Another major concern in this paper is that Poulter (2013, 2017) viewpoint on citizenship in Finnish RE seems to lack a critical perspective. Although her studies were based on educational policy documents, she did not concentrate on the specific thematic content of the selected educational policy documents relating to citizenship in RE in Finland. Rather, she dealt with the conception of what an ideal citizen is in different epochs in Finnish RE history. The study also treated the ideological trends defining citizenship in Finnish RE. She drew on national core curricula for basic education and didactical manuals of Lutheran RE for about 150 years (1860s–2000s). She observed that citizenship in Finnish RE has shifted from an institutional religious orientation to that of individual religious

conviction: a shift from the traditional communal image of obedient Lutheran Finns to personal religious convictions. (Note that this kind of thought relating to individual religious conviction, unreservedly upheld in Poulter (2013, 2017), has been the subject of criticism by some scholars (e.g., Sakaranaho 2013).) (See more about this in the latter part of this paper.)

Poulter claimed that being a good Lutheran is no longer a precondition for being a good Finn. The previous idea of commitment to the national (Lutheran) church is being replaced with ideas of self-realization, personal aspiration and individually conceived ethical responsibility (Poulter 2017). According to her, this curricular development is underpinned by and/or can be analysed in terms of governmentality, liberal individualism, secularization and post-secularization (Poulter 2017, pp. 187–92, 198–200). Her study was based on content analysis (Poulter 2017, p. 189), which is generally descriptive and of realist assumptions (Hardy et al. 2004, p. 21; Schreier 2012, p. 47). Realist assumptions in this case seem to relate to empirical (descriptive) realism rather than critical realism, as Poulter's work does not challenge the content of the policy documents she investigated. Similarly, her research is of empirical realism because she unreservedly upholds her positions relating to personal religious conviction and citizenship in Finnish RE (cf. Bryman 2004, p. 12). Moreover, Poulter (2017) never claimed to have employed critical realism.

Adebayo and Mansikka (2018) examined the security concerns relating to citizenship in Finnish RE. Their study employed some education-cum-security policy documents of Finland in relation to some selected United Nations (UN) policy documents. Their findings indicated that Finland is subtly employing citizenship in RE to address some human rights- and religion-related personal security concerns that are characteristically related to human security as conceptualised by the UN. Finland seeks to ensure freedom from fear (an element of human security) for Finns, as many Finns see immigrants as threats to the sustenance of their national Christian religious (loosely rendered cultural) heritage and identity. Additionally, Finland seeks to ensure freedom from fear for immigrants, as they are faced with the possibility of being assimilated—culturally and/or religiously.

Following Adebayo and Mansikka (2018), Finland seeks to address the security concerns by: (a) allowing the individual pupil (both Finn and immigrant) to receive RE and citizenship in RE in his or her own religion so as to allay fear relating to assimilation and loss of religious and cultural identity and heritage; (b) enabling Finns and immigrants to mutually learn and understand each other's cultures and religions with a view to enhancing good ethnic relations; (c) incorporating global citizenship, human rights, and "critical thinking" ideas into its new RE curriculum with a view to making young people critical consumers of information. This aims at preventing them from being radicalised through extremist ideologies and religious fundamentalism.

Unlike Poulter's studies, Adebayo and Mansikka (2018) seem to have treated some specific thematic content of the national core curricula and of the other selected policy documents relating to citizenship in Finnish RE. However, their analysis (like that of Poulter) was of qualitative content analysis (QCA) and realist assumptions. Hence, it also lacks critical insight, as it is simply a description of what is actually contained in the examined documents.[3] In this instance, their descriptive-oriented approach does not give room to examine whether or how the thematic content (such as human rights issues) presented in the policy documents is defective or detrimental to the pupils (cf. Schreier 2012, p. 47; Hardy et al. 2004, p. 21). Briefly, there is a need for a critical perspective on the subject in question so as to see whether there is any lapse in the relevant government policy.

Therefore, this research focuses on critical approaches using maximal interpretation of citizenship education as its theoretical framework and critical discourse analysis as its method of analysis. This paper examines some thematic content of selected policy texts with a view to searching for maximal citizenship in Finnish RE. It investigates the relationship between the 21st-century national core

---

[3]    Note that Adebayo and Mansikka (2018) specifically acknowledge that their work is not critical. Hence, their realist assumptions automatically favour empirical realism rather than critical realism (Bryman 2004, p. 12).

curricula for basic education and the other selected policy documents of Finland. This study focuses on grades 7–9 of basic education[4] in relation to the school subject under investigation. The main finding and argument of this work is that citizenship in Finnish RE is only somewhat compatible with the characteristics of maximal citizenship. This is due to the inconsistencies and inadequate explicit connections between democratic, secular, religious and educational values in the selected documents. This signals adverse effects on the potential for critical-mindedness of young people and equal access to human rights education. Meanwhile, the study shows how a Finnish linguistic conception of "citizenship" in relation to the recent development in national educational policy pushes the conception of maximal citizenship in a relatively new direction. Moreover, it reveals how an explicit use of the "Convention on the Rights of the Child" in Finnish curriculum broadens our conception of maximal citizenship in general.

### 1.3. Research Questions

The investigation seeks to address the following questions:

1. How compatible are the elements of citizenship in RE in the selected Finnish national core curricula for basic education with maximal citizenship?
2. How do the recent developments concerning citizenship and RE in the selected Finnish policy documents interrelate in view of maximal interpretation of citizenship education?
3. How could the compatibility and developmental issues raised in questions 1 and 2 above affect the pupils (young people) at the level of education in view?

### 1.4. Structure of the Paper

The next section discusses maximal interpretation of education for citizenship as the theoretical framework for this inquiry. It subsequently gives a background relating to "Finnishness" and diversity in today's Finland with a view to reinforcing the need for critical views on citizenship in Finnish RE. It thereafter states the research design and methods of this study. This is followed by the analysis of the empirical data derived from the selected policy documents. The article subsequently gives a summary of its major findings, a further reflection on the analysis of the policies and a discussion on how this work attempts to advance the discourse on maximal citizenship beyond the context under investigation. This study concludes by stating its limitations and future research directions.

## 2. Theoretical Framework

### Maximal Interpretation of Education for Citizenship

In 1992, McLaughlin introduced the theory of maximal interpretation with a view to dealing with some philosophical difficulties relating to citizenship education. He conceived of maximal interpretation of citizenship as an alternative to the minimal interpretation of citizenship (McLaughlin 1992; Miedema and Avest 2011). McLaughlin's maximal interpretation of education for citizenship remains relevant in this 21st century as much as it was in 1992 (McLaughlin 1992; Fouts and Lee 2005, p. 35). Besides the above, we will see further (below) that there is currently considerable support for a more maximal citizenship education in Europe and strong research evidence about "maximal" approaches from Europe and the USA seem to lend credence to this (Jackson 2007, pp. 30–32; cf. Fouts and Lee 2005).

Meanwhile, the minimalist interpretation visualises citizenship and its related identity merely in formal, legal and juridical terms. It presents citizenship education as a knowledge-based subject with a

---

4　Young people between the ages of 13 and 15 are usually expected to be in grades 7–9 of Finnish basic education. Note that Finnish basic education is compulsory, and it begins when the child turns seven. (See section 25 of the Basic Education Act 628/1998; Amendments up to 1136/2010 in (Finland 2010).

specific civics-related content to be transmitted in a formal and didactic manner. Hence, a citizen is understood as someone having a certain civil status, with its associated rights and responsibilities, within a community based on the rule of law. In other words, a citizen should be law abiding and "public spirited", as loyalties and responsibilities are mainly seen as local and immediate in character (McLaughlin 1992, p. 236; Fouts and Lee 2005, p. 45; Miedema and Bertram-Troost 2014, p. 75). Maximal citizenship, on the other hand, is couched in cultural, social and psychological terms. Maximal views require a more extensive focus for their loyalty and responsibility. Possessing a passport or having the right to vote is not enough for maximal citizenship. It entails consciousness regarding obligations, rights, common good and fraternity, among other things, in a shared democratic and universal context. The maximal interpretation strives for continuous redefinition of citizenship. For maximalists, critical understanding and perpetual questioning is crucial in citizenship education (McLaughlin 1992, pp. 236–37).

Nonetheless, minimalist and maximalist interpretations are not necessarily opposite to each other, but form a continuum. Perhaps the most salient contrast between minimal and maximal views for educational purposes concerns the degree of critical understanding and questioning that is regarded as necessary for citizenship (McLaughlin 1992, pp. 236–40). In a maximal sense, one could be a responsible and law-abiding citizen and simultaneously be a critical and an active citizen in pursuit of social justice and empowerment of all citizens. Accordingly, maximal citizenship is the critical development of minimal citizenship (Fouts and Lee 2005, pp. 44–45). This perhaps led DeJaeghere to suggest that a "critical approach" should replace the term "maximal" as the ideal for citizenship education. For her, this critical approach would help provide the necessary conditions for collective social change through a combined focus on "knowledge" and "participation" (Johnson and Morris 2010, p. 86). However, replacing the term maximal with critical approach could be misleading at times, as not all studies (on citizenship education) that claim to be critical would agree with maximal viewpoints. Macedo (2003), for example, claims to be critical but he is averse to inclusion of religious beliefs in public and liberal citizenship education. This contradicts McLaughlin (1992) maximal conception.

Kerr (Kerr 1999; Johnson and Morris 2010, p. 85) proposes that McLaughlin (1992) minimal-maximal model should be used to differentiate between civic education and citizenship education. Civic education, in Kerr's proposal, means education for minimal citizenship while citizenship education means education for maximal citizenship. However, Johnson and Morris (2010) have pointed out that many countries use subject labels that do not concur with this type of interpretation. For example, France uses the term *éducation civique* (civic education), which does not necessarily suggest minimal interpretation of citizenship (Johnson and Morris 2010, p. 85). Therefore, the preference for citizenship education rather than civic education in this paper should not be understood as reflecting the minimal (civic as uncritical) and maximal (citizenship as critical) distinction in Kerr's proposal.

Johnson and Morris (2010) claim that Westheimer and Kahne (2004) and Veugelers (2007) complicate the minimal–maximal distinction by dividing citizens into three types. For Westheimer and Kahne (2004), these are: personally responsible citizen, participatory citizen and justice-oriented citizen. For Veugelers (2007), they are: adapting citizen (that emphasizes transfer of values and norms), individualistic citizen (that emphasizes individual choice and resultant responsibility) and critical-democratic citizen (that promotes autonomy and critical inquiry about all knowledge and values). (Note that neither Westheimer and Kahne (2004) nor Veugelers (2007) raised any complications or correlations between their three types of citizens and minimal-maximal citizenship.)

The take in this paper is that the relationship between the three respective types of citizens (mentioned just above) and minimal/maximal citizens is not necessarily complicated. This seems possible provided we treat the notion of "citizen" in each model as a continuum rather than exclusively distinct. For instance, Westheimer and Kahne (2004) personally responsible citizen (i.e., obedient, honest and law-abiding citizen) appears to be similar to minimal citizenship. Moreover, a combination of Westheimer and Kahne (2004) participatory citizen (i.e., social, problem-solving and politically

informed citizen) and justice-oriented citizen (i.e., social, problem-solving, critical and questioning citizen that seeks to change established systems/structures that perpetuate injustice over time) seems similar to maximal citizenship. This appears logical, as one needs to be well informed to appropriately engage in critical questioning in pursuit of justice. To reinforce the continuum, we need to note that a justice-oriented citizen would not necessarily encourage a personally responsible citizen to break the law.

Meanwhile, McLaughlin (1992) minimal-maximal citizenship stands out, as it seems to strike a balance in treating the relationship between religious beliefs and liberal ideologies. It is unlike Westheimer and Kahne (2004) model of citizenship that is generally indifferent in this respect. It is also unlike that of Veugelers (2007) that somewhat downplays the significance of religion in matters of education for personal identity and moral development in denominational (faith) and state schools.

McLaughlin (1992) applies the minimalist-maximalist continuum to the issue of state neutrality in matters of religion—advocated in liberalism. He argues that the state, although it must be neutral in matters of private good (like religious views), has a non-neutral commitment regarding the fundamental principles of justice affecting the notion of the good in public realms. On such a basis, the state could realise equilibrium between diversity and cohesiveness (McLaughlin 1992, p. 240). Hence, citizenship education offered by the state should include religious views within the maximalist perspectives (cf. McLaughlin 1992, p. 240ff). McLaughlin (1992) focused on an interpretation of citizenship education that could be an account of public civic virtues that is substantial enough to satisfy the communal demands of citizenship, yet compatible with liberal demands concerning the development of critical rationality by citizens. He hoped his account will satisfy the demands of justice relating to diversity (McLaughlin 1992).

As noted above, Jackson (2007, pp. 30–32) has indicated that the research projects on citizenship (sponsored by the CoE and the EC) bear out the difference between McLaughlin's minimal and maximal interpretations of citizenship education. According to him (i.e., Jackson), the findings of the EC/Eurydice project, for instance, recommend that citizenship education should be detached as far as possible from its legal connotation with a view to embracing all members of a given society, regardless of their nationality, sex, race, social or educational status. In this instance, there is widespread support for citizenship education's role in developing political literacy by employing democracy and human rights issues with a view to increasing the active participation of the pupils. Furthermore, the EC/Eurydice report recommends that citizenship education could be offered as a separate subject or be integrated into conventional subjects—such as religious and moral education. (Note that similar views are expressed in relation to the findings of the CoE project—EDC (Jackson 2007, pp. 30, 39)). However, the CoE project (EDC) did not directly treat religion as an aspect of citizenship education (Jackson 2007, p. 40). Given that minimal-oriented citizenship promotes unreflective socialization, the EC/Eurydice report seems to promote maximal-oriented citizenship that emphasizes inclusion and active and interactive learning that is values-based and process-led (Jackson 2007; EC/Eurydice 2017, p. 24).

The essence of mobilizing for maximal capacities in relation to the citizenship curriculum is to view teaching as a practice of persistence and hope that acts against the inequitable status quo and to contest, resist and transform injustice (Keddie 2008, p. 182). In order to realise the maximal conception of citizenship, the citizens need to have a considerable degree of explicit understanding of democratic principles, values and procedures. This should be combined with the dispositions and capacities required for participation in democratic citizenship, generously conceived (McLaughlin 1992, p. 237). The term "explicit" in this sense should not be misconstrued as delineating deterministic tenets of citizenship that young people must learn and imbibe as does Dieltiens (2005).

Dieltiens critiques maximal citizenship for seeking to fill in the details of what is required of learners to be ideal citizens of a diverse and liberal democratic society. She argues that the proponents of maximal citizenship have, in this respect, slipped too far from a major purpose of education (i.e., the development of autonomous individuals). She states further that her "quarrel" with maximal

citizenship is its attempt to "coerce" young people to accept a set of deterministic liberal tenets (Dieltiens 2005). Dieltiens (2005) seems to misrepresent maximal conception because McLaughlin (1992) states that maximal citizenship is "dynamic" but not "static". That something is "dynamic" means it is amendable (understandably). Any educational curriculum for citizenship is continually subject to questions, redefinitions and reforms as far as maximal citizenship is concerned (McLaughlin 1992). As such, associating deterministic tenets with maximal citizenship seems misleading. Moreover, McLaughlin (1992) and other maximal educators never suggested the use of "coerce" in promoting maximal citizenship. Therefore, "coercing" young people to conform with certain moral values is not part of maximal citizenship. The term "explicit" in relation to maximal citizenship is meant for "explicit understanding" of democratic ethos and public virtues (based on national debate, dialogue and consensus) that a society may contain in its policy (e.g., national curriculum). The reality is that human society needs "explicit" guidelines (curriculum) in matters of education for citizenship, given its complexity and contentiousness. Such guidelines are necessary to ensure equal opportunities and to avoid chaos or trivialization of important civic themes. (The analysis and discussion in this study demonstrate this.)

Following the above, it seems the minimal interpretation of citizenship education is open to objections. Notable among these is its potential to promote merely an unreflective socialization into the political and social status quo. It is therefore inadequate for educational purposes (McLaughlin 1992, p. 238). The maximal approach to citizenship education, despite its critical elements, is equally not immune to objections. Owing to the controversial questions that maximal interpretation may open up, it is in danger of presupposing a substantive set of "public virtues" that may exceed the principled consensus that exists or can be achieved (McLaughlin 1992, p. 241). This objection is insignificant because an education that lacks controversial content is certainly unsatisfactory. Such an education cannot develop autonomous persons (Callan 1997). Controversial questions are needed to delineate citizenship education and a resolution that may arise from the ensued controversies (cf. McLaughlin 1992, p. 245). Hence, McLaughlin (1992, p. 238) finds it reasonable to support maximal citizenship education. This owes much to the fact that it requires a much fuller educational programme whereby the development of a broad critical understanding and a much more extensive range of dispositions and virtues relating to a general liberal and political education are seen as crucial. He also supports the maximal conception because it requires the consideration of a more explicit egalitarian thrust in educational arrangements (Fouts and Lee 2005, p. 46).

Maximal citizenship seems relevant to this study for the following reasons: (1) the maximal approach to citizenship education is values-based (Jackson 2007, p. 32; Miedema and Bertram-Troost 2014, p. 75) just as RE is values-oriented. Thus, the two are combinable; (2) McLaughlin advises that citizenship education in the state schools should entail religious viewpoints within the framework of the maximal approach (cf. McLaughlin 1992, p. 240ff). In a similar manner, an EC project on citizenship that bears out the maximal approach notes that citizenship education can be integrated into RE (Jackson 2007, pp. 31–32). Accordingly, the ideas of the maximal approach and citizenship in RE seem interwoven and compatible; (3) a major aim of RE in Finland is to help pupils develop their "critical" potential (FNBE 2014, p. 435). This objective is characteristic of maximal citizenship education (McLaughlin 1992, pp. 237–38). Hence, this paper holds that maximal interpretation of citizenship education can help us critically understand the civic values in Finnish RE.

## 3. Finnishness and Diversity: A Need for Critical Views on Citizenship in Finnish RE

Piattoeva (2009, 2010b) gives some glimpses of the need to engage in critical perspectives on citizenship-related studies in Finland even though her studies lack explicit discussion on maximal interpretation. Piattoeva (2009) claims that Finnish citizenship and what traditionally constitutes "Finnishness" were recently scrutinised. She states further that the traditional conception (beginning in the mid-19th century) of Finnish citizenship related to the ethno-nationalistic Finnish word *kansalaisuus* (citizenship) began to be questioned in the 1980s. (See more about the word *kansalaisuus* vis-à-vis

another ethno-nationalistic Finnish word *kansalainen* (citizen) in the policy analysis section below.) The scrutiny challenges the traditional representation of Finland as a homogenous nation-state (Piattoeva 2009, pp. 726–27). This is partly owing to the steady influx of foreigners of diverse cultural and religious backgrounds and the entry of Finland into the CoE (in 1989) and the European Union (in 1995) (cf. Piattoeva 2009, pp. 726–27; Piattoeva 2010a, p. 69; Raento and Husso 2001).

Piattoeva (2010b) investigated political projects that impact the objectives, content and methods utilised in Finnish schools. She notes that recent government multicultural initiatives are "in many ways immune ... to multicultural [and multi-religious] thinking" (Piattoeva 2010b, p. 8). As such, she advises that "public education should acknowledge and break away from its nationalistic and ethnocentric past, its original task to forge national unity at the expense of diversity" (Piattoeva 2010b, p. 7).

As noted earlier, the ideas behind individual religious convictions such as those unreservedly upheld in Poulter (2013, 2017) have been a subject of criticism by some scholars (e.g., Sakaranaho 2013; Scheinin as cited in Sakaranaho 2013, p. 244). Although their critiques are in relation to Finnish RE in general, their works seem implicitly instructive to see citizenship in Finnish RE in a critical perspective. The crux of their critiques deals with the current legal provision about Finnish RE that makes Lutheran RE compulsory for Lutheran pupils (the majority). (Notable here are the Finnish Freedom of Religion Act 2003 and section 13 of the Basic Education Act 628/1998 [Amendments up to 1136/2010] (Sakaranaho 2013)). In contrast, the same legal provision does not make the RE of the minority religions (apart from the Orthodox) compulsory for the pupils of the minority religions. In this arrangement, the pupils of the minority religions could opt for the Lutheran RE, but the Lutheran pupils cannot opt for the RE of the minority religions. Besides, there must be at least three minority pupils of a given religion (registered in Finland) backed by parental requests before the RE for such a religion can be arranged. The legally required parental requests (from minority pupils) is however not applicable to the Lutheran and Orthodox RE. The critiques indicate that the exclusive privileges given to the Lutheran and Orthodox forms of RE is because the Lutheran and Orthodox churches are legally the national churches of Finland (Sakaranaho 2013; Scheinin as cited in Sakaranaho 2013, p. 244; Ubani 2013, p. 207).

Furthermore, the critiques raise some questions about freedom/equality/human rights. The inconsistency in the legal provision appears as placing the state institutional religious interest (i.e., the two national churches' interest) above the personal religious conviction of individual citizens in matters of choice in Finnish RE (Sakaranaho 2013; Scheinin as cited in Sakaranaho 2013, p. 244). Evans (2001) seems to share similar sentiments as she is critical of the European Court of Human Rights and the European Commission of Human Rights for interpreting and declaring this kind of legal provision as upholding freedom of religion and belief. Nonetheless, the legal provision under scrutiny subsists to date.

Juxtaposing the above critiques with Poulter (2017), one could see that some positions maintained therein can be questioned. Notably, her view stating that citizenship in Finnish RE has shifted from an institutional religious orientation to that of individual religious conviction seems short of critical perspective. The issue is that Poulter took for granted the legal compulsion or restriction influencing the pupils' choice of RE, which ultimately determines the kind of citizenship in RE they will receive. That being a good Lutheran is no longer a pre-condition for being a good Finn does not suffice here, as individual religious convictions do not require compulsion or restriction in favour of or at the expense of other faiths/beliefs.

Similarly, Poulter (2017) and Adebayo and Mansikka (2018) position stating that the Finnish policy allows individual pupils to receive RE and/or citizenship in RE in his or her own religion is short of critical insight as it takes for granted the fact that there are legal restrictions that somewhat affect this.

## 4. Research Design and Methods

### 4.1. Case Study

Case study is the research design (Bryman 2004, p. 33; Gerring 2007) chosen for this work. Case study entails the detailed, intensive and complex analysis of the case under investigation (Bryman 2004, pp. 48–49). This article examines maximal citizenship in Finnish RE as a case study. The purpose is to make a detailed and intensive analysis of the case subject (Bryman 2004, p. 48) and to treat it as a national-bound case among European nations (cf. Gerring 2007, p. 20). In part, this Finnish case seeks to increase our knowledge about maximal citizenship in RE among nations and democracies in Europe (cf. Gerring 2007, p. 20). The choice of RE in this paper is to serve as an example of values-education-oriented school subjects (such as social studies, history and ethics among others) into which general citizenship education could be integrated (Jackson 2007, p. 31). The idea of maximal citizenship in Finnish RE in this case research seeks to exemplify maximal perspectives relating to values education in democratic contexts in general.[5] Notably, this inquiry takes RE in Finland as an "exemplifying case" (cf. Bryman 2004, p. 51) in Europe, as it shows how the elements of maximal citizenship are being used in an educational curriculum of a European nation (cf. Jackson 2007, pp. 30–32).

Furthermore, the case of Finnish RE in this study is significant, as maximal citizenship has not been examined in relation to the RE of any Nordic country (in Northern Europe). As such, the maximal perspective in this inquiry is about Finland as a Nordic and European case. The case research seeks to give us valuable insights into the unique model of Finnish RE in terms of the maximal conception. The uniqueness of Finnish RE entails the fact that it allows each pupil to receive RE according to his/her own religion provided certain requirements are met (as discussed above). This is not the case with RE of the other Nordic countries. For instance, Sweden provides a single model of RE for every pupil regardless of their religion (Flensner 2015), i.e., mono-religious education. A similar model of RE is found in Norway (Skeie 2003) and Denmark (Buchardt 2014).[6]

### 4.2. Sources of Data

The sources of empirical data in this research are Finnish policy documents. Specifically, these are: the new (2014) and the earlier (2004) NCCBE authored by the Finnish National Board of Education—FNBE (FNBE 2004, 2014). The NCCBE is the national framework on which local and school curricula must be based (FNBE 2004, 2014); *Global Education 2010*—this document is authored by the Ministry of Education (MoE) and it is a collection of national development objectives and measures needed to realise global compliant education for citizenship (MoE 2007); the 2004 and 2009 government reports (prepared by the Ministry for Foreign Affairs—MFA) about the Human Rights Policy of Finland (MFA 2004, 2009)—the two reports contain information about human rights education (MFA 2004, 2009) that is deemed relevant to citizenship and human rights issues in Finnish RE; the government resolution (document) regarding future migration in Finland—this text is from the Ministry of the Interior (MoI) and it contains views of Finns about internationalization and migration (MoI 2013) that seem to call for global education for citizenship; and last is a government resolution about internal security (also authored by the MoI) containing some views of Finns about immigrants that are relevant civic values (MoI 2008).

---

5   The distinctive characteristics of the "critical" elements in maximal citizenship will be discussed in more detail below.
6   The approach to mono-religious education offered in each of these countries (i.e., Sweden, Norway and Denmark) has its unique elements. However, the comparison and details of such uniqueness are not within the purview of this case study. (See below for more details on the "Limitations and Future Research Directions" of this study).

*4.3. Method of Analysis*

A critical form of discourse analysis is the method of analysis employed in this research. Critical discourse analysis usually criticises the values transported by the dominant discourse in favour of the less powerful and the underprivileged (Schreier 2012, pp. 46–47, 50; cf. Patton 2002, pp. 98–100).

Generally, discourse analysis is done by analysing the language in texts with a view to tracing elements of discourses (Baker 2006, p. 5). The goal of discourse analysis in all its forms (descriptive or critical) is to analyse the ways whereby language contributes to the construction of social reality. It assumes that language itself is not reality (Schreier 2012, pp. 45–46). Thus, discourse analysts are interested in both how language is used and how it is not used in constructing social reality. In effect, they usually analyse what is and what is not in the materials that may be under investigation (Schreier 2012, p. 47). This means discourse analysis (critical or descriptive) is based on constructivist assumptions (Schreier 2012, pp. 46–47, 50; cf. Patton 2002, pp. 96–100). In constructivism, social phenomena and their meanings are continually being accomplished by social actors. Thus, knowledge is treated as indeterminate, as researchers can only present a possible but not the exact version of social reality (Bryman 2004, p. 17; cf. Baker 2006, p. 4).

Considering the above, critical discourse analysis seems to be useful or relevant to this study (at least) for the following four reasons: first, the critical principles promoted in critical discourse analysis enable this work to examine the subject under investigation from a critical perspective. Hence, the analysis in this paper is a departure from the previous studies about citizenship in Finnish RE which are generally uncritical. Second, the critical-oriented approach that critical discourse analysis upholds appears to align with the critical orientations embedded in maximal citizenship—the model of citizenship in focus. Third, this study deals (in part) with how the linguistic conception of citizenship in the selected documents reflects a construction of social reality. This correlates with the fact that discourse analysis (in general) is interested in how language contributes to the construction of social reality. Fourth, similar to the focus of discourse analysis in general, this research is less interested in what is being said but more in how things are being said about the phenomena under investigation (Schreier 2012, p. 48). A QCA approach would have been appropriate if the research interest were the other way around (Schreier 2012, p. 48; Adebayo and Mansikka 2018).

Critical discourse analysis is used in this research in the following ways: (1) this article analyses a set of linguistic categories relating to citizenship in RE in Finnish policy. (2) This research seeks the thoughts that are explicitly and implicitly expressed in the selected documents that may or not reflect the language of maximal interpretations of citizenship education. In other words, the paper discusses how citizenship in Finnish RE is compatible and how it is incompatible with the characteristics of maximal citizenship. The discourse thus reveals the strengths and weaknesses of citizenship in Finnish RE in terms of the maximal conception. (3) The study is critical of the government policy on RE that could be detrimental to the maximal citizenship of young people. (4) It analyses some recent policy-oriented strategies that Finland employs in attempting to impact the social world of young people. The aim here is to see how effective and ineffective the government strategies could be. (5) Owing to the constructivist-assumptions-inspired discourse in this study, the interpretation made of the selected policy texts is couched as a possible rather than an absolute one.

## 5. Analysis of Policy Documents Regarding Maximal Citizenship in Finnish RE

The analysis (results) in this work is based on the following categorizations: (a) global citizenship, ethno-nationalism and linguistic turn, (b) the issues with human rights education and (c) on critical thinking, reflective judgment and dialogic education. The categorizations are not exclusive. They are simply different perspectives on the questions in focus.

### 5.1. Global Citizenship, Ethno-Nationalism and Linguistic Turn

The Finnish MoE has noted that citizenship skills that transcend national borders are important and needed virtues in modern society (MoE 2007, p. 9). This is partly informed by the steady influx of migrants and the resulting effects in Finland since the 1990s.

> As immigration began to accelerate in the early nineties, internationalisation began to be more and more tangibly felt in the streets in Finland, too. Conflicts between Finns and immigrants appeared in the headlines. There was a growing need for fostering tolerance and preventing discrimination. It became the task of global education to enhance intercultural understanding . . . and to foster awareness of one's prejudices and change attitudes. (MoE 2007, p. 9)

Following another publication of the MoE, the development calls for education that emphasizes global responsibility through global education (Kaivola and Melén-Paaso 2007). Global education is conceived as the global dimension of citizenship education (Kaivola 2007, p. 3). MoE (2007, p. 14) thus demanded that "[g]lobal education must be included in the [national] core curricula of all forms and levels of education . . . when they are next reviewed". The directive envisages that the "local and school curricula will define the role of global education in different subjects and indicate the subjects of particular importance in this respect" (MoE 2007, p. 14). Thus, the new NCCBE (FNBE 2014, p. 19) gives preference to "[g]lobal education as part of basic education [that] contributes to creating preconditions for fair and sustainable development in line with UN development goals".

Global citizenship, as required by the MoE, has been incorporated into the new RE curriculum. "The instruction of religion supports the pupil's growth into a responsible member of his or her community and the democratic society as well as a global citizen" (FNBE 2014, p. 435). The explicit mention of "global citizen" as an objective of RE in the new NCCBE (FNBE 2014) represents a shift in paradigm about the conception of Finnish citizenship for at least two reasons. First, there was no explicit reference to "global citizen[ship]" as a goal of Finnish RE in the previous national curriculum. Citizenship in Finnish RE in the curriculum from 2004 is inferred from a cross-curriculum theme, "[p]articipatory citizenship and entrepreneurship" (FNBE 2004, p. 38). Questions concerning citizenship could also be deduced from the ethical issues relating to Finnish RE in the NCCBE, 2004 (Poulter 2017). Second, employing RE to support every pupil becoming a responsible global citizen (rather than just a responsible citizen) seems to counter the ethno-nationalistic connotation in the Finnish word, *kansalainen*, for citizen. The most appropriate linguistic conception for *kansalainen* would be a "national" rather than a "citizen" or "global citizen" (cf. Piattoeva 2009, pp. 726, 740).

The Finnish words *kansalainen* (translated citizen), *kansalaisuus* (translated citizenship), *kansallisuus* (translated nationality) and *kansakunta* (translated nationhood) etymologically have a common root word *kansa* meaning nation (cf. Piattoeva 2009, pp. 726, 740). Hence, the Finnish word for citizen linguistically and conceptually forecloses global dimensions of citizenship altogether. It rather portends absolute nationalism. As such, the Finnish traditional conception of citizenship seems to be far from global ideals, as it emphasizes ethno-nationalism rather than universality of humanity.

Introducing "global citizen" (*maailmankansalainen*) into the 2014 NCCBE is a major departure from the traditional linguistic and conceptual norm. *Maailmankansalainen* (a global citizen) would be, linguistically, appropriately conceptualised as a "global national". It thus seems to neutralise the disequilibrium in the traditional Finnish conception of citizenship, as it could somewhat convey global and national dimensions of citizenship in native speakers' minds. The idea of global citizenship, rather than ethnic-oriented and nationalistic-oriented citizenship, in the new RE curriculum emphasizes social justice as a consequence of growing diversity. This is in line with "[t]he social task of basic education [which] is to promote equity, equality and justice" (FNBE 2014, p. 19). This educational task, dealing with equity, equality and justice, is fundamental to the principles of maximal citizenship in general (McLaughlin 1992; Miedema and Bertram-Troost 2014).

Accordingly, the linguistic paradigm shift (above) makes a central value of maximal interpretation of education for citizenship visible. The curricular transformation of ethno-nationalistic citizenship in favour of global citizenship seems to demonstrate that Finland is advancing maximal citizenship. In this instance, maximal citizenship is critical of the previously normal ethno-nationalistic citizenship and renders it inappropriate in an increasingly diverse Finland. This development seems compatible with the minimal-maximal continuum of citizenship, and it could help satisfy the demands of justice relating to diversity and religion-related values in education (cf. McLaughlin 1992, p. 235; Miedema and Bertram-Troost 2014, p. 75). The analysis above indicates how a linguistic development relating to the conception of "citizenship" in RE could contribute to the construction of social reality (cf. Bryman 2004; Schreier 2012).

Nevertheless, the idea of "global citizenship" needs to be more explicitly integrated into Finnish RE, particularly in connection with the human rights content of its curriculum. The idea of global citizenship appearing just once in the new Finnish RE curriculum (FNBE 2014, p. 435) seems inadequate, as it could be taken for granted in practice. This is premised on the fact that the ethno-nationalistic Finnish word for and conception of citizenship that emphasizes nationality/ethnicity rather than the generality of humanity has long been entrenched in Finnish society (Piattoeva 2010a, pp. 66–69). This appears more pertinent as many Finns still view internationalization and migrants negatively in favour of nationalism:

> Finland has had relatively little experience as a host country for migrants [and immigrants], and this is perhaps one explanation for the dominance of rather negative views of migration among Finns, whereby internationalisation and migration have been seen as a threat to national culture. (MoI 2013, p. 9)

Ironically, "those who view immigrants with a negative attitude do not perceive their opinions as racism but as 'cautious wisdom'" (MoI 2008, p. 9; Adebayo and Mansikka 2018). Rather than taking such "cautious wisdom" for granted, it needs scrutiny in view of human rights and global citizenship because extreme nationalism is now a matter of concern in Europe. Accordingly, the inadequacy relating to the explicit connection between the linguistic undertones of global citizenship and human rights issues is inimical to maximal citizenship. The term "explicit" is a matter of necessity in this case, as "[m]aximal conceptions require a considerable degree of explicit understanding of democratic principles" (cf. McLaughlin 1992, p. 237) relating to human rights and civic values (Miedema and Bertram-Troost 2014).

### 5.2. The Issues with Human Rights Education

For the first time, human rights issues are explicitly integrated into citizenship in Finnish RE in the new NCCBE (FNBE 2014, pp. 435–42). It is noteworthy that human rights had not been explicitly mentioned in connection with citizenship in Finnish RE in the earlier NCCBE (FNBE 2004). The new development is somewhat in line with the EC and the CoE projects on citizenship that see human rights issues as part of citizenship education (Jackson 2007, pp. 30–31). Following the new curriculum, teachings on ethics in RE should not just be teacher-pupil discussions. They should equally cover the interrelations between religious ethics and human rights as enjoined by the UN:

> Current and societal ethical questions are discussed in the teaching and learning. These include ecosocial knowledge and ability . . . and the role of religions in building peace in the society and the world. The ethics of the studied religion and other religions and worldviews as well as the UN Universal Declaration of Human Rights are emphasised in the contents. Human rights violations . . . are also examined. (FNBE 2014, p. 437)

The teaching on religious ethics in relation to the UN Universal Declaration of Human Rights and human rights ethics is specifically mentioned in Lutheran, Orthodox, Islamic and Judaic forms of RE in the new NCCBE. This is found under the heading "Good life" of the four categories of RE (FNBE 2014,

pp. 439–42). The Orthodox RE seems to be unique among the four forms of RE, as it is the only one that includes the UN Convention on the Rights of the Child.

However, the idea of human rights issues appears to be downplayed in Catholic RE. The issue here is that the concept of human rights is not explicitly and specifically connected to it at all. One could only make a connection between Catholic RE and human rights issues by making inference from the human rights and ethical concepts mentioned in the general introduction to the RE curriculum (FNBE 2014, pp. 435–37). The non-inclusion of human rights issues in Catholic RE seems to require critical scrutiny. It suggests that Catholic RE is less open to secular/liberal ethics (Šimenc 2003, p. 3) in comparison with other forms of RE. This also means that citizenship in Catholic RE falls somewhat short of the maximal viewpoint because it somewhat satisfies conservative religious demands of citizenship at the expense of the liberal demands relating to the development of critical rationality by citizens (cf. McLaughlin 1992, p. 235). Succinctly put, this could ill equip the pupils of Catholic RE as to how religious perspectives are perceived within the secular and liberal cycle. It could also make them deficient in matters of the relevant "religious responses to liberal and secular values" (Pike 2012, p. 111, cf. pp. 117–18). As such, realizing a balanced perspective in matters of religious and secular values for the pupils of Catholic RE seems to be at stake.

But for Catholic RE, the linguistic undertones of citizenship in Finnish RE are moving beyond the rhetoric of religious tolerance and mutual respect to a more concrete and explicit discourse on human rights. The reasoning behind this is to develop the pupils into responsible, democratic and global citizens (FNBE 2014, p. 435). This development suggests an emerging "tripartite intertwinement" of democratic citizenship education, worldview education (including religious view) and human rights education in Finnish RE. Similar to Miedema and Bertram-Troost (2014), this paper notes that this "tripartite intertwinement" could be strengthened by the principles of maximal citizenship education.

Nevertheless, the fact that only the curriculum of Orthodox RE includes the UN Convention on the Rights of the Child seems to raise another critical question in the sense that: "[t]eaching pupils about the UN Convention on the Rights of the Child is part of human rights education. [However, t]he content and the obligatory nature of the Convention are not ... well known in Finland" (MFA 2009, p. 97). Hence, the Finnish government recommends that "the Convention must be incorporated into the curricula of schools" (MFA 2009, pp. 97–98). The concern here is that the incorporation subsequent to this recommendation could lead to unequal opportunities, as the Convention has only been incorporated into the Orthodox RE curriculum, not the other forms of RE. It seems this kind of curricular arrangement is somewhat incompatible with and could be disadvantageous to realizing maximal citizenship. The reasoning here is that such an arrangement is unlikely to promote a much fuller and broader critical understanding of the Convention relating to the rights of the child among the pupils of other forms of RE. Moreover, it has the potential to diminish a broader range of dispositions and virtues relating to a general liberal and political education and a more explicit egalitarian thrust in educational arrangements. All these are also crucial to a maximal conception (McLaughlin 1992, p. 238).

Integrating the UN Convention on the Rights of the Child into the other forms of RE is significant, as the rights of the children to maximally participate in a democracy is contingent upon it. Finland notes that:

> *The right of the child to participate* is a question of children's right to express their opinion and be genuinely heard in all matters concerning them according to their age and level of maturity, of children's right to seek, obtain and supply information, and of children's right to express themselves. (MFA 2009, p. 29) (emphasis original)

Finland specifically requires its "municipalities to ensure that children and adolescents can influence matters and are heard in decisions concerning them" (MFA 2009, p. 165). The Finnish government notes further that a crucial factor in promoting children's and young people's rights "to participate is to ensure that, as required under human rights conventions, the education received

by children and young people reinforces their individual creativity and social responsibility" (MFA 2009, p. 29). The UN Convention on the Rights of the Child is said to be relevant here (MFA 2009, pp. 29, 165). As such, it appears that the non-inclusion of the Convention in every (but Orthodox) form of RE in Finland could negatively impact many children and young people in expressing their opinions and in being genuinely heard in matters pertaining to religious phenomena and their rights and responsibilities. One may inquire whether children could effectively participate in a matter that they are not adequately knowledgeable about. This development is inconsistent with the FNBE recommending "that the rights of the child [should] be . . . strengthened in the preparation of local curriculum" (MFA 2004, p. 162). Meanwhile, the national curriculum bearing such inconsistencies is the official framework for preparing local curricula (FNBE 2014, p. 3).

The above suggests that the lack of explicit connection of human rights issues with Catholic RE and the non-inclusion of the Convention on the Rights of the Child in every (but the Orthodox) form of RE seem not to be in accord with the new curriculum that seeks common objectives in all forms of RE. This is articulated thus: "[i]n order to ensure the coherence of the subject, common objectives and content areas have been determined for all forms of instruction in the pupil's own religion" (FNBE 2014, p. 439). This kind of curricular contradiction seems to reinforce the possibility of inequality in matters of critical understanding, creativity ability, and sense of responsibility relating to religion-related values and rights among the young people. Hence, the RE curriculum in this respect also falls short of elements of maximal citizenship.

### 5.3. On Critical Thinking, Reflective Judgment and Dialogic Education

Critical thinking and reflective judgment are major aims of the new Finnish RE curriculum. It states that "[t]he pupils are guided towards critical thinking and the observation of religions and worldviews from different viewpoints" (FNBE 2014, p. 435). The new NCCBE categorically connects personal reflective judgment to ethical questions in RE.

> In teaching and learning, the pupils familiarise themselves with ethical thinking in the studied religion and in other religions, and they are encouraged in personal reflection on ethical questions . . . The instruction provides the pupil with elements for building and evaluating his or her . . . personal view of life and worldview. (FNBE 2014, p. 435; cf. FNBE 2004, p. 202)

The intended role of teachers in the NCCBE is not to promote specific religious ideologies but religious ethics in general. The teachers' task is to make room for discussions on religious ethics and different worldviews with the pupils in order to encourage them to develop independent judgments. "When implementing the syllabi . . . joint situations promoting learning, participation, interaction and dialogue skills are created . . . and the [pupils'] ability to take initiative are guided and reinforced" (FNBE 2014, p. 437).

The foregoing reveals that citizenship in Finnish RE promotes a potential for analysis, criticism, self-reflection, actorship and authorship of the students in line with general liberal and political education (cf. Miedema and Avest 2011, p. 414). This seems to concur with the maximal interpretation of citizenship education, as the new curriculum seeks to enhance critical citizenship in RE rather than subjecting the children to uncritical socialization (cf. Castelli and Trevathan 2008, pp. 87–88; Miedema n.d.). The new curriculum puts it more explicitly: "[e]ducation shall not demand or lead to religious . . . commitment of the pupils" (FNBE 2014, p. 16). "[T]he aim is here the students' self-responsible religious self-determination" (Miedema and Avest 2011, p. 414).

The dialogic idea in the new RE curriculum appears as a potential resource to enhance critical reflection and creative power in individual pupils. This seems so, as a dialogic approach would not see pupils as passive beings (cf. McLaughlin 1992, p. 237) that must unquestionably learn some specific civic values in matters of religion. It is rather a means of negotiation and creation of a space where learning and encounters of others' worldviews can occur. In this case, the dialogic approach could afford the learners opportunities to seek critical understanding and continual questioning that are seen

as necessary to education for citizenship relating to religion. All these are generally related to maximal interpretation of citizenship (cf. McLaughlin 1992, p. 237; Miedema and Avest 2011).

Meanwhile, dialogic education may remain a mere wish as there are no clear-cut dialogic guidelines in the new RE curriculum. The claim that "[t]he instruction of religion supports the pupils' ability to participate in the dialogue within and between religions and worldviews" (FNBE 2014, p. 435) seems inadequate. This claim itself is a sequel to an earlier policy direction ("Dialogue between religions and world views") of the MoE, stating that "cognitive, comparative content of different religions will be accommodated into the teaching of religious studies ... in general education" (MoE 2007, p. 14). The aim relating to the enhancement of dialogue between religions appears more obscure because dialogue in the maximal approach should entail an interreligious character (Miedema and Avest 2011, pp. 417, 422). Meanwhile, the present model of Finnish RE (though non-confessional) does not bring pupils of different religious backgrounds together in the same class. Hence, it seems there is a need to develop specific and explicit dialogic curricular guidelines regarding citizenship in the subsisting mode of RE if the maximal potentials of dialogic education are to be maximally realised. Otherwise, critical thinking and reflective judgment ideals may not be fully maximised.

## 6. Summary, Discussion and Advancement

A significant finding about maximal citizenship in Finnish RE entails a linguistic turn relating to the concept of global citizenship. Through its policy documents, Finland seems to be seeking to whittle down the dominance of the Finnish nationalistic and ethno-centric word (*kansalainen*) for citizen in its RE curriculum. It does this by introducing (explicitly) the idea of global citizen (*maailmankansalainen*) into its new national RE curriculum (FNBE 2014) as a departure from the previous curriculum (FNBE 2004). This development suggests a progression along the minimal-maximal continuum regarding the conception of citizenship in RE, as it now treats its traditional conception as not unquestionable (McLaughlin 1992).

The linguistic-related findings in this work suggest that a consideration of the meaning of the term "citizenship" in people's languages (spoken and written) could further advance our conception of maximal citizenship. This seems to challenge previous studies about maximal citizenship, as they are devoid of linguistic analyses of "citizenship". Scholars have for a long time restricted studies about maximal citizenship mainly to policymaking or curricular matters in schools. Yet, language is arguably an early instrument that shapes many peoples' conception of citizenship in life. This development in turn determines whether such a conception will be minimal or maximal as well as its subsequent impact on the making and implementation of policy or curriculum. Hence, this article suggests that the traditional linguistic conception of citizenship and its subsequent development (if any) among all peoples could be a better starting point to study minimal and maximal citizenship in general. The foregoing suggests that all education policy makers need to be cautious of possible "minimal" elements in the linguistic conception of citizenship in their mother tongues with a view to ensuring maximal citizenship in policymaking.

Nevertheless, the findings reveal that there is a need to ensure that the concept of global citizenship is made more explicitly visible in connection with human rights issues in the Finnish RE curriculum. This is of critical importance in citizenship education, as global citizenship would always find its relevance in human rights. Indeed, it is the universality of humanity (not nationality) in human rights issues that makes global citizenship global. Despite its unifying nomenclature, the reality is that the global world is not a country yet. This is affecting how countries (including Finland) operate.

For a long time, Finland has politically projected itself as a homogeneous nation-state though the country has always had its indigenous ethnic minorities (Piattoeva 2010a, p. 66): Swedish-speaking Finns, Romani and Sami (cf. Adebayo and Mansikka 2018). Perhaps this is the reason why the Finns who have a negative attitude towards immigrants do not see their views as racism (MoI 2008). Owing to recent increases in diversity, this seems no longer sustainable. There is now a need to enhance social cohesion among Finns, immigrants and migrants. Aptly, the Finnish government acknowledges that

Finns are relatively inexperienced in hosting or living with people of foreign descent (MoI 2013). As such, elements of global citizenship that are more visibly connected to human rights issues with a view to prioritizing humanity rather than ethnicity are more needed than ever in Finnish RE curricula. The government now needs to make a more explicit curriculum that would enable young people to query the issues of a negative attitude towards immigrants and racism within the context of religion and global citizenship. The need to be aware one's prejudices and negative attitudes towards others should be unambiguously stated in the NCCBE. All these are necessary because the traditional and ethno-nationalistic conception of citizenship has long remained dominant in the minds of many Finns (Piattoeva 2010a). Otherwise, the feasibility of global citizenship relating to the conception of maximal citizenship could be stunted.

This case research somewhat advances the discourse on maximal citizenship in relation to human rights. This seems so, as this paper reveals the need for explicit mentioning of the UN Convention on the Rights of the Child in Finnish RE curriculum in terms of maximal citizenship. This is significant, as the relevant previous studies loosely reveal the relationship between human rights, children's rights, maximal citizenship and RE. The relevant previous studies do not explicitly mention the UN Convention on the Rights of the Child that corresponds to the young people's age. As such, liberal (secular) values and ethics could now be more concretely and critically combined with those of religion in education. This is more in line with maximal education for citizenship, as the maximal conception continually seeks tenable ways of combining liberal/secular and religious values (McLaughlin 1992; Miedema and Avest 2011; Miedema and Bertram-Troost 2014).

The introduction of human rights issues into the new RE curriculum (FNBE 2014) could help raise some thought-provoking questions about the relationship between human rights issues and religious phenomena for learning and teaching maximal citizenship. On this note, one needs to concur with a thought-provoking statement stating that RE is intrinsically a human right and that religious beliefs and human rights phenomena are inseparable (Schweitzer 2016). A case in point here is child marriage, a practice that, on account of religious beliefs, was legally permitted in Finland until May 2019. It is noteworthy that the country's immediately past Justice Minister, Antti Häkkänen,[7] had called for a child marriage ban in Finland as in the other Nordic countries (*Yle* 2018). Subsequently, child marriage is now legally banned in Finland as of June 1, 2019. The aim is to protect the rights of children (*Yle* 2019). While this may be commendable, such a law or policy is not good enough if young people are not educated regarding the ban and the reason underlying it. It seems any education that engages young people in issues of this type is likely to influence their reasoning and attitudes towards human rights, religion and related social life issues (cf. Ziebertz 2016). Therefore, it could be productive to engage young people in matters of this kind in relation to their rights, religious practices and beliefs about marriage and maximal citizenship in educational contexts.

A major deficiency found in the national curriculum is that human rights issues are not specifically and explicitly incorporated into Catholic RE. In addition, the UN Convention on the Rights of the Child is only mentioned in Orthodox Finnish RE (FNBE 2014). As such, all (but Orthodox) RE pupils may not be able to learn about the intersection between the Convention and religion until the next curriculum reform (usually after a 10-year interval). This appears to marginalise a large section of young people (RE students) across the country. The lapse seems more serious, as the Convention's content and obligatory nature is not well known in Finland (MFA 2009). The take here is that researchers and policy makers in general should take the issues of the UN Convention on the Rights of the Child in relation to RE seriously, as such correspond to young people's age. Furthermore, any citizenship in RE (Finnish or non-Finnish) that seeks to be maximal needs to engage with liberal values much as it would with religious values. Otherwise, RE pupils would be one-sidedly educated in matters of

---

[7]   He was the country's Minister of Justice from early May 2017 to early June 2019.



secular and "sacred" citizenship education. This seems detrimental to participatory democracy, as the potential of maximal citizenship education may not be maximised.

Moreover, the idea of critical thinking and personal reflective judgment about ethics and religion in the national RE curriculum could be far from feasible since it fails to equip the children with human rights instruments that adequately correspond to their age. Indeed, it is impossible for anyone to think critically or to make appropriate judgments about any rights he or she is not aware of. It is understood that the idea of dialogue in the Finnish RE curriculum does not entail explicit and specific dialogic guidelines. As such, dialogic education relating to maximal citizenship in Finnish RE policy may not adequately translate to practice. By extension, this means explicit dialogic guidelines are prerequisites to advancing maximal citizenship in general.

The use of the word "explicitly" or "explicit" in relation to global citizenship, human rights, the Convention on the Rights of the Child and dialogic issues in this research is a matter of necessity. This is because "[m]aximal conceptions require a considerable degree of explicit understanding of democratic principles, values and procedures on the part of the citizen, together with the dispositions and capacities required for participation in democratic citizenship generously conceived" (McLaughlin 1992, p. 237).

Contrasting the view of Adebayo and Mansikka (2018), this study suggests that incorporation of human rights issues, ideas of the global citizen and critical thinking in explicit terms into the new RE curriculum does not necessarily mean that Finnish RE now promotes "critical global citizenship" and equality. Moreover, the potential relational values between human rights, the global citizen, critical thinking and RE do not simply make it so. As one can see, subsisting Finnish educational policy is incapable of promoting adequate critical-mindedness, critical global citizenship and equal opportunities for human rights education among young people. Therefore, researchers do not have to unquestionably accept "critical" claims that may be contained in any educational policy. Following the principles of maximal education, the policy shortcomings discussed in this paper should be subject to continual national debate, dialogue and consensus with a view to ensuring maximal citizenship in educational policymaking.

Despite its limitations, the notion of maximal citizenship as concerns Finland seems related to the concept of a good citizen noted in an American-based study.[8] The study concludes that a "good citizen" is one who cares about the welfare of others, as they deal with others in moral and ethical contexts. It also notes that a good citizen challenges and critically questions ideas, proposals and suggestions, and can make good choices based upon good judgment, in light of existing circumstances (Fouts and Lee 2005, p. 32). This relational perspective further reinforces the fact that questioning and critical citizenship (maximal) does not necessarily negate responsible, law-abiding and ethics-oriented citizenship (minimal).

Nevertheless, the idea of a "critical" approach relating to citizenship education does not necessarily mean a "maximal" approach. Specifically, any critical approach dealing with citizenship education that sees liberal education (based on an interpretation of liberalism) and religious beliefs as mutually exclusive seems to contradict the tenets of the "maximal" approach. Briefly put, the critical perspective in maximal citizenship is not indifferent to religious beliefs. It also does not marginalise religious beliefs as seen in some "critical" approaches to citizenship education. (See examples above.) This "maximal" study reinforces the fact that religion remains pivotal to the notion of citizenship or citizenship education among many private citizens and public authorities across the world (cf. Fouts and Lee 2005, pp. 27–30). Accordingly, this paper challenges the literary theory (McLaughlin 1992) that unreservedly treats every "critical" approach to citizenship as "maximal". (See above.) This should be considered when one may be searching for maximal citizenship in educational policy in any context.

---

8    The idea of good citizenship could vary from one person/society/country to another (cf. Fouts and Lee 2005, pp. 32–33).

Considering the discussion about the empirical findings in this research, the answers to the three specific research questions stated at the beginning of this paper are briefly highlighted as follows:

On research question 1 (i.e., How compatible are the elements of citizenship in RE in the selected Finnish national core curricula for basic education with maximal citizenship?), this work suggests that citizenship in Finnish RE is only somewhat compatible with the elements of maximal citizenship. The compatibility limitation is mainly due to the inconsistencies and inadequate explicit connections between democratic, secular, religious and educational values in the selected country's curricula. More specifically, the national curricular shortcomings concern global citizenship ideas and language, children's and human rights issues, the dialogic idea, religious ethics and RE. As such, the elements of maximal citizenship in Finnish RE are inadequate. This is not to deny that a part of the compatible elements pushes the conception of maximal citizenship in a relatively new direction.

On research question 2 (i.e., How do the recent developments concerning citizenship and RE in the selected Finnish policy documents interrelate in view of maximal interpretation of citizenship education?), the study reveals that the recent developments interrelate by the nexus of their thematic content which is inherently related to the maximal conception. Notably, the idea and language of global citizenship, the UN Universal Declaration of Human Rights and the UN Convention on the Rights of the Child issues recently and explicitly linked to RE in the Finnish policies are interrelated in terms of maximal citizenship. These policy developments interrelate (I) as they aim to satisfy both liberal and religious communities in matters of citizenship in RE and (II) as they seek to blend liberal and religious values together with a view to raising critical citizens and promoting justice in an increasingly diverse Finland. Such goals are essentially maximal in character. This is evident in the fact that the central aim of maximal interpretation of education for citizenship is to find common ground between liberal and communal (including religious community) demands with a view to building critical citizens and satisfying the demands of justice in a diverse society (McLaughlin 1992). Meanwhile, the highlighted interrelation is not strong enough as a result of the shortcomings enumerated against research question 1 above.

On research question 3 (i.e., How could the compatibility and developmental issues raised in questions 1 and 2 above affect the pupils [young people] at the level of education in view?), the paper suggests that the compatibility and developmental issues raised in questions 1 and 2 could positively but inadequately impact young people. This could lead to inefficiency in teaching related to maximal citizenship in Finnish RE, a possible scenario that can negatively affect the pupils. This portends adverse effects on the potential for critical-mindedness of young people and equal access to human rights education. This could also ill-equip young people to engage in participatory democracy as Finland advocates. Nevertheless, the positive aspect of the raised compatibility and developmental issues could positively impact the maximal capability of the pupils but only to some extent.

## 7. Limitations and Future Research Directions

While there are common values between religion and culture, the idea of diversity relating to RE in this work is not necessarily about multicultural education. The critical viewpoint relating to the theoretical framework that guides this research is restricted to maximal interpretation of citizenship education. It does not deal with the relationship between the critical elements of maximal citizenship and the theories of critical thinking (cf. Johnson and Morris 2010), critical pedagogy (cf. Johnson and Morris 2010; Miedema and Bertram-Troost 2014), cosmopolitanism (cf. Waghid 2014), and their cognates. This study is primarily interested in the maximal interpretation of citizenship education propounded by McLaughlin (1992). The paper illuminates McLaughlin's maximal interpretation of citizenship with other major scholars' works on citizenship but mainly only those that have built on it and attempted to "complicate", rejig, critique and promote it from the 1990s up to the 21st century.

This paper restricts the discussion on the legal compulsion and restriction (mentioned above) affecting citizenship in Finnish RE to secondary literature. The secondary literature seems to have critically examined the relevant legislative documents containing the legal compulsion and restriction

regarding Finnish RE in general. Thus, this article simply reviews them with a view to suggesting the descriptive nature of the previous studies about citizenship in Finnish RE. This study does not necessarily reflect the actual policy implementation and experience at the classroom and individual levels, as it is based on national policy documents.

The idea of citizenship in this inquiry is understood as an integrated subject within the "separate" and "distinct" subject of RE. This article is not against RE as a "separate" and "distinct" subject in favour of a general citizenship education in school.

This is a single case study. It is primarily about Finnish RE. It does not combine several cases of the RE of different countries in relation to the maximal conception of citizenship. Its primary maximal focus is that of Finland. As such, it does not primarily employ a "comparative method" in analysing maximal citizenship in RE of different countries.[9] The emphasis on comparison of RE in this study deals with the relevant matters of analysis relating to the different forms of RE contained in the Finnish curriculum. This is meant to maintain a focus on the research questions of this work. Meanwhile, the paper ensures that the focus on Finnish RE in relation to the concept of maximal citizenship for young people is consistently argued, justified and analysed within a broad context of international scholarly literature. Accordingly, this study contains a detailed, complex and an intensive analysis of the case under investigation.

Given that the search for maximal citizenship in educational policy in this inquiry is in the Finnish case, this study could be transposed to the contexts that may be lacking this type of research. Such contexts could be RE and other relevant school subjects in other countries. This transposed idea could enable us to understand the elements of maximal citizenship peculiar to individual school subjects and different educational, democratic and national contexts. This seems tenable, as McLaughlin originally proposed that his theory of maximal citizenship for educational curricula should apply to diverse contexts and democratic societies in general (McLaughlin 1992).

**Funding:** The Article Processing Charge (APC) was funded by the University of Helsinki, Helsinki, Finland.

**Acknowledgments:** I appreciate the efforts of Mari Laukka, Outi Ala-Kahrakuusi, Pirita Seitamaa-Hakkarainen and the editors of this journal on this article. I am very grateful to Robert Whiting for his comments. The reviewers of this paper are also appreciated for their resourceful review reports and comments.

**Conflicts of Interest:** The author declares no conflict of interest.

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
