# Peer review of "In Search of Maximal Citizenship in Educational Policy for Young People: Analysing Citizenship in Finnish Religious Education in View of the “Maximal” Conception"

_socsci, doi:10.3390/socsci8080232_

Round 1
Reviewer 1 Report
Please reconsider the structure of the article.
Please answer the three research questions (as mentioned in the beginning) at the end of the article.
Further comments are written in the text.

Author Response
"Please see the attachment."

Reviewer 2 Report
I find your work valuable. I have learned with this work something outside the spectrum of my specific interests in the field of religious education. I think you have done a good job of research and more work and scholarly lectures can be derived from this article.
Author Response
"Please see the attachment."

Reviewer 3 Report
This is surely an interesting topic of research, and your article is well structured and mostly well argued. I like the way you showed a shift between Finnish regulation from 2004 to 2014 and the clear change regarding the idea of a global citizen. It is also good that you contradict to other authors who claims that a simple change of the language and the introduction of the issues like human rights, global citizen and critical thinking in the RE curriculum will promote the idea of "critical global citizenship" and equality.
However, I do have two major concerns. First of which, the author(s) does not explain why this is important topic of research. Theoretical rationale of the article is not the problem, but why Finnish religious education is important? Is it somehow different from other Scandinavian countries, and if so, how? Or is it completely unique? If it is unique, how and why?
That brings me to my second concern. Such article has to use a comparative method. I would be happy to read a comparative study of different RE, as it will strenghten the validity of results.
Author Response
"Please see the attachment."
